# Conversion of *Escherichia coli* into Mixotrophic CO_2_ Assimilation with Malate and Hydrogen Based on Recombinant Expression of 2-Oxoglutarate:Ferredoxin Oxidoreductase Using Adaptive Laboratory Evolution

**DOI:** 10.3390/microorganisms11020253

**Published:** 2023-01-19

**Authors:** Yu-Chen Cheng, Wei-Han Huang, Shou-Chen Lo, Eugene Huang, En-Pei Isabel Chiang, Chieh-Chen Huang, Ya-Tang Yang

**Affiliations:** 1Department of Electrical Engineering, National Tsing Hua University, Hsinchu 30013, Taiwan; 2Department of Life Sciences, National Chung Hsing University, Taichung 40227, Taiwan; 3Department of Food Science and Biotechnology, National Chung Hsing University, Taichung 40227, Taiwan; 4Program in Microbial Genomics, National Chung Hsing University, Taichung 40227, Taiwan; 5Innovation and Development Center of Sustainable Agriculture, National Chung Hsing University, Taichung 40227, Taiwan; 6Program in Microbial Genomics, Academia Sinica, Taipei 115, Taiwan

**Keywords:** carbon fixation, reductive tricarboxylic acid cycle, 2-oxoglutarate:ferredoxin oxidoreductase, adaptive laboratory evolution

## Abstract

We report the mixotrophic growth of *Escherichia coli* based on recombinant 2-oxoglutarate:ferredoxin oxidoreductase (OGOR) to assimilate CO_2_ using malate as an auxiliary carbon source and hydrogen as an energy source. We employ a long-term (~184 days) two-stage adaptive evolution to convert heterotrophic *E. coli* into mixotrophic *E. coli*. In the first stage of evolution with serine, diauxic growth emerges as a prominent feature. At the end of the second stage of evolution with malate, the strain exhibits mixotrophy with CO_2_ as an essential substrate for growth. We expect this work will open new possibilities in the utilization of OGOR for microbial CO_2_ assimilation and future hydrogen-based electro-microbial conversion.

## 1. Introduction

Global warming has brought attention to the need to mitigate atmospheric CO_2_ concentration [1]. Among all strategies, CO_2_ conversion is regarded as a prominent route to creating an alternative pathway to transform CO_2_ into biomass and value-added chemicals as the ultimate carbon sink. There has been extensive work on CO_2_ assimilation based on the Calvin cycle and the ribulose monophosphate (RuMP) cycle [2,3,4,5,6,7]. Recently, Gleizer and coworkers converted heterotrophic *Escherichia coli* into autotrophic *E. coli* with adaptive laboratory evolution using formate as an energy source [5]. The basic rationale behind their approach was as follows: the heterologous expression of non-native enzymes expands the space of possible metabolic reactions, but this alone will not guarantee the needed flux will flow through the newly expanded set of reactions. The approach combines rewiring of the central metabolism to establish a dependence on newly added carboxylation flux and provide a selective advantage to utilize the autotrophic pathways. Various adaptive laboratory evolution techniques such as batch, chemostat, turbidostat, and morbidostat have been reported to achieve novel phenotypes and shed light on the interplay of selective pressure, historical contingency, and epistatic effects [4,5,7,8,9,10,11]. In this work, we attempted to use the 2-oxoglutarate:ferredoxin oxidoreductase (OGOR) in the reductive tricarboxylic acid (rTCA) cycle as the carboxylation enzyme for carbon fixation (Figure 1a). The rTCA cycle was first discovered in 1966 in *C. thiosulfatophilum* by Aronon, Buchanan, and coworkers and has been extensively studied more recently [12,13,14,15,16,17,18,19,20,21]. The rTCA cycle is less energy consuming than the Calvin cycle, involving enzymes such as OGOR that are oxygen-sensitive, and it is therefore found in anaerobes or microaerophilic bacteria [15,16,17,18]. OGOR fixes CO_2_ to succinyl-CoA, forming 2-oxoglutarate and CoA (coenzyme A) using ferredoxin as an electron donor. Previously, we have pointed out the pivotal role of OGOR in establishing in silico hydrogen-powered autotrophy in *E. coli* based on constraint-based modeling with flux balance analysis [22]. Moreover, the CO_2_ assimilation is also confirmed with *E. coli* overexpressing ATP citrate lyase and OGOR with glucose as both the carbon source and the energy source [23]. Here, we would like to establish mixotrophical CO_2_ assimilation solely based on heterologous OGOR (Figure 1a) using hydrogen as an energy source in *E. coli*. *E. coli* is known to possess innate hydrogenase, and growth on carbon sources such as malate with hydrogen has been demonstrated [24].

## 2. Materials and Methods

### 2.1. Strains and Genomic Modification

The 2-oxoglutarate:ferredoxin oxidoreductase (OGOR) was cloned from an anaerobic phototrophic bacterium *Chlorobaculum tepidum strain* TLS genome (GenBank accession number: AE006470) [23]. To construct an expression vector containing the genes of two sub-units of OGOR, *korAB* (CT0163 and CT0162 gene symbol in GenBank, total 2.9 kb), were amplified from cellular DNA from *C. tepidum* strain TLS (ATCC 49652/DSM 12025/NBRC 103806) as gene cassettes with a designed primer pair by Ordered Gene Assembly in *Bacillus subtilis* (OGAB) method (Table 1). The designed DNA cassettes were cloned into plasmid pGETS118, and the plasmid was extracted and transformed into the regular *E. coli* cloning host strain BW25113 via heat-shock transformation. The plasmid was also re-sequenced to confirm *korAB*. The 2-oxoglutarate:ferredoxin oxidoreductase (OGOR) was cloned from an anaerobic phototrophic bacterium Chlorobaculum tepidum strain TLS genome (GenBank accession number: AE006470) [23]. To construct an expression vector containing the genes of two sub-units of OGOR, the korAB (CT0163 and CT0162 gene symbol in GenBank, total 2.9 kb) were amplified from plasmid pGETS-KAFS [23] as a gene cassette with a designed primer pair by Ordered Gene Assembly in Bacillus subtilis (OGAB) method [25]. The primer pair used in cloning experiments is shown in Table 1. The designed DNA cassette was cloned into plasmid pGETS118 and driven by a pR promoter [26]. The plasmid was extracted and transformed into the regular *E. coli* cloning host strain BW25113 via heat-shock transformation. The plasmid was also sequenced before and after evolution experiments with next-generation sequencing, and the sequences of korAB genes were the same as pGETS-KAFS with a frameshift mutation (717 deleted A) in korA [23].

### 2.2. Enzymatic Activity Assays of 2-Oxoglutarate:Ferredoxin Oxidoreductase

The *E. coli* strains were incubated in M9 medium supplied with 2 mM MgSO_4_, 0.1 mM CaCl_2_, 10 mM sodium nitrate, 0.01 mM NiCl_2_, 0.01 mM FeCl_2_, 50 μg/mL thiamine, and 20 μg/mL chloramphenicol under anaerobic conditions. To obtain anaerobic conditions, H_2_ and CO_2_ were used to purge the headspace of a bottle sealed with rubber. Approximately 4 × 10^9^ bacteria cells were collected by centrifugation (10,000× *g*, at 4 °C) from 750 mL medium and sonicated in Tris buffer (100 mM Tris-HCl at pH 8.4, 3 mM dithioerythritol) on ice in an anaerobic chamber (Coy Laboratory Products, Inc., Grass Lake, MI, USA) filled with 9% H_2_, 15% CO_2_, and 76% N_2_ gas. The crude extracted proteins were collected from the supernatant by centrifugation (10,000× *g*, at 4 °C). The 2-oxoglutarate:ferredoxin oxidoreductase activity assays were performed using the method described in a previous report with modifications [15] in an anaerobic chamber (Coy Laboratory Products, Inc., Grass Lake, MI, USA). The activity of 2-oxoglutarate:ferredoxin oxidoreductase was determined by the reduction in succinyl-CoA (reduced methyl viologen:succinyl-CoA oxidoreductase). The assay was performed in a 1 mL volume containing 100 mM Tris-HCl at pH 8.4, 2 mM MgCl_2_, 4 mM methyl viologen, and 1 mM succinyl-CoA. Dithionite was added from 1 M stock solutions until the methyl viologen-containing assay solutions presented a faint blue color. The optical density changes were measured at 578 nm (methyl viologen, ε_578_ = 9.8 × 10^3^ M^–1^ cm^–1^) after addition of the succinyl-CoA solutions. The optical densities were measured with a GeneQuant 1300 (GE Healthcare, Little Chalfont, Buckinghamshire, UK) in the anaerobic chamber at 35 °C. The result is displayed in Table 2.

### 2.3. Growth Condition and Data Processing

The pre-culture *E. coli* strains were inoculated into M9 minimal medium with 2 mM MgSO_4_, and 0.1 mM CaCl_2_, 0.01 mM NiCl_2_, 0.01 mM FeCl_2_, and 10 mM NaNO_3_. The medium was supplemented with 20 μg/mL chloramphenicol and sodium malate or L-serine as carbon sources. To generate and keep the anaerobic culture environment at an elevated CO_2_ concentration during the evolution, we used a commercial anaerobic gas pack (AnaeroPack-Anaero, Mitsubishi Gas Chemical Inc., Tokyo, Japan) and airtight box (Mitsubishi Gas Chemical, Tokyo, Japan) of volume 2.5 L and inner dimensions 13.5 cm × 19.7 cm × 9.5 cm. Alternatively, for post-evolution analysis, we instead used an anaerobic bag (Mitsubishi Gas Chemical, Tokyo, Japan) with N_2_ purge for no CO_2_ conditions and an anaerobic gas pack. For the cultivation involving hydrogen, we used a hydrolysis chemical reaction of sodium borohydride in strong sulfuric acid to generate hydrogen gas, i.e.,
NaBH_4_ + 2H_2_ O → NaBO_2_ + 4H_2_

We first prepared NaBH_4_ powder (0.3 g in weight) in a gelatin capsule and placed the capsulate in strong sulfuric acid (~18 M in concentration and 60 mL in volume) [27,28]. The gelatin capsule dissolves in sulfuric acid after approximately 2 min, which allowed us to have enough time to close the lid. The cell mixing is performed simultaneously with gas bubbling and magnetic stirring [28]. The voltage reading data with a temporal resolution of 12 min from the phototransistor was recorded in an Arduino microcontroller board (Arduino UNO, Arduino Inc., Scarmagno, Italy) and wirelessly transmitted to the external computer. After conversion of the voltage reading to optical density with a calibration curve obtained from a plate reader (Synergy H1 Hybrid; Biotek, Winooski, VE, USA), the growth curve was extracted via a parameter-free code in Python 3 [29]. For post-evolution analysis, we thawed and pre-cultured the cells in serine containing minimal medium for 1 day. Subsequently, the clone was transferred to a bioreactor in an anaerobic bag with desired gaseous conditions under passive diffusion. The active pumping used in evolution resulted in slightly different growth characteristics from analysis obtained with passive diffusion, but the difference was not central to the analysis of the result. For all the growth experiments, triplicate runs were performed unless otherwise stated.

### 2.4. Evolution in the Bioreactor

In short, ten milliliter cultures were grown for ~48 hr for each passage at 37 °C. (Each growth cycle, cells grow 100-fold, corresponding to log_2_ 100 = 6.6 generation.) After ~48 hr growth, cells were diluted 1:100 into a tube with fresh medium to start another passage. Meanwhile, samples were frozen (−80 °C) with glycerol for storage for post-evolution analysis. For the evolution experiment, the hydrogen gas was actively pumped into the culture vial via a micro diaphragm pump (T2-05 VBIC, Parker Hannifih Inc., Cleveland, OH, USA) and a syringe filter.

### 2.5. Next-Generation Sequencing and Transcriptome Analysis

#### 2.5.1. Genomic DNA Isolation and Sequencing

Isogenic bacterial cells were grown for 48 hr with 0.6 g/L malate as carbon sources and H_2_ supply. The *E. coli* strains were harvested, washed with 1X PBS, and resuspended with 400 μL GT lysis buffer (RBC Bioscience, New Taipei City, Taiwan) containing a mixture of 100 μm/400 μm silica beads (OPS Diagnostics, Lebanon, NJ, USA). The bacterial cell wall was disrupted by a bead-milling process, and DNA was subsequently purified by MagCore Genomic DNA Tissue kit (RBC Bioscience, New Taipei City, Taiwan) according to the manufacturer’s protocol. Purified DNA was measured with a ND-1000 spectrophotometer for the purity (Thermo Fisher Scientific, Waltham, MA, USA), a Qubit 4.0 fluorometer (Thermo Fisher Scientific, Waltham, MA, USA) with Quant-iT dsDNA BR Assay kit for the quantitation, and analyzed with the Agilent 4200 TapeStation (Agilent Technologies, Santa Clara, CA, USA) with the Genomic DNA ScreenTape assay for the DNA integrity check. For libraries, we used the Illumina Truseq DNA library kit. Finally, all samples were sequenced on an Illumina Sequencer using the 150PE protocol.

#### 2.5.2. Genomic DNA Variant Analysis

The data then went through quality control (QC) and were subsequently aligned to a reference genome of BW25113 in the database using the Burrows–Wheeler Alignment tool (BWA). Afterwards, variant calling and subsequent annotations were created by the Genome Analysis Toolkit (GATK) [30] and SnpEff [31].

#### 2.5.3. RNA Isolation, Library Preparation, and Sequencing

Total RNA was isolated from *E. coli* strains that were grown for 48 hr with 0.6 g/L malate as carbon sources and H_2_ supply. Total RNA was extracted using Trizol^®^ Reagent (Invitrogen, Carlsbad, CA, USA) according to the instruction manual. Purified RNA was quantified at an optical density (O.D.) of 260 nm using a ND-1000 spectrophotometer (Nanodrop Technology, Waltham, MA, USA) and qualitated using a Bioanalyzer 2100 (Agilent Technology, Santa Clara, CA, USA) with the RNA 6000 LabChip kit (Agilent Technology, Wilmington, DE, USA). All RNA sample preparation procedures were carried out according to the Illumina’s official protocol. The SureSelect XT HS2 mRNA Library Preparation kit (Agilent, Santa Clara, CA, USA) was used for library construction, followed by AMPure XP beads’ (Beckman Coulter, Brea, CA, USA) size selection. The sequence was determined using Illumina’s sequencing-by-synthesis (SBS) technology (Illumina, San Diego, CA, USA). Sequencing data (FASTQ reads) were generated using Welgene Biotech’s pipeline based on Illumina’s basecalling program bcl2fastq v2.20. Differential expression analysis was performed using StringTie (StringTir v2.1.4) and DEseq (DEseq v1.39.0) or DEseq2 (DEseq2 v1.28.1) with genome bias detection/correction and Welgene Biotech’s in-house pipeline.

## 3. Results

Based on a previously reported experiment on RuBisCO, we decide to use adaptive laboratory evolution (ALE) as a metabolic optimization tool [4,5,27]. We employed a custom anaerobic bioreactor, specifically designed for robust anaerobic cultivation with hydrogen (Figure 1b) [28]. A long-term evolution experiment can be carried out with ease. High temporal resolution (~12 min) growth data enable the calculation of the growth rate from a statistics-based non-parametric method to infer the first-time derivative from time series data [29]. This technique is powerful, as it provides high temporal resolution growth data to reveal fine features, such as diauxic growth, to identify the novel phenotype. The 2-oxoglutarate:ferredoxin oxidoreductase were cloned from an anaerobic phototrophic bacterium *Chlorobaculum tepidum strain* TLS. The OGOR enzyme activity assays were performed using a method described in a previous report with modifications (Table 2) [23]. After the transformation of the strain, the strain did not assimilate CO_2_, and we used serine and malate as auxiliary carbon sources to establish basal-level growth for evolution. Serine is known to be preferably co-metabolized and a transporter of serine, which allows for its efficient transport [30,31]. Malate was chosen because *E. coli* is able to utilize malate as carbon source only when H_2_ is available [24]. For serine or malate as a sole carbon source, we first established that H_2_ either rescues or enhances microbial growth of the ancestral strain before commencing the evolution experiment (Appendix A). During the evolution, we allowed for the active diffusion of H_2_ and CO_2_ into the sample, and hence, mutants with CO_2_ assimilation capability will have selective advantage over repeats of competition. In total, three replicate evolution runs were carried out, and only one yielded the mixotrophic strain at the end of evolution. The result of the evolution that yields the mixotrophic strain is presented in detail.

We first evolved the strain in serine at a constant concentration of 0.5 g/L and an initial growth rate of 0.06 hr^−1^ (Figure 2a,b). On the 26th day (~91 generations), diauxic growth emerged as a predominant feature, and the growth rate of the second phase (secondary growth mode) was ~0.007 hr^−1^ (Figure 2c). In addition, the growth rate of the first phase (primary growth mode) had a significant increment from 0.087 hr^−1^ (26th day) to 0.14 hr^−1^ (28th day). Henceforth, we designate the first phase in diauxic growth as primary and the second phase as secondary unless otherwise stated. At this point, we decided to switch to malate as the auxiliary carbon source and evolve the strain for the second stage with a progressively decreasing malate concentration. The diauxic growth reappeared in the initial phase of the second stage of evolution with an initial malate concentration of ~4.85 g/L, and therefore, diauxic growth is not specific to serine. The growth rates of both the first and second phases increased as the evolution went on in the first 20 days (Figure 3a,b). When the malate concentration was reduced to ~12.5% of the initial malate concentration, or equivalently ~0.6 g/L, on the 40th day in the second stage, the diauxic growth behavior disappeared, and growth was single phase (Figure 3c). The corresponding growth curves before and after the disappearance of diauxic growth are shown in Appendix A. (Indeed, the critical disappearance of diauxic growth is repeatable in two “replay” evolution experiments of the subclone from the 34th day in the second stage.) We continued to evolve the strain until the 144th day. The concentration was as low as ~1% of the initial malate concentration (~0.05 g/L). (From the 112th day onward, at ~3% of the initial malate concentration, the growth was still detectable, but the data could not be reliably processed for the growth rate calculation).

At the end of long-term (~184 days in total) two-stage (40 days in serine and 144 days in malate) evolution, frozen samples were thawed and checked. We checked the evolved clone from the second stage at a malate concentration of 1.2 g/L. Under elevated CO_2_ conditions (pCO_2_ = 0.2 atm), the growth rate was ~0.144 ± 0.02 hr^−1^ (mean± sd, *n* = 3), whereas no growth was detected for the N_2_ purge with H_2_ supplied, i.e., no CO_2_ but with H_2_ supplied (Figure 3d). The dependence of the growth of the evolved clone on CO_2_ availability shows CO_2_ assimilation. More precisely, CO_2_ is an essential substrate for growth. In contrast, the ancestral and wild type showed an inferior growth rate at elevated CO_2_ conditions with H_2_ supplied, possibly due to CO_2_-induced stress (Appendix A) [32]. For additional control experiments, we also profiled the ancestral strain and wild-type BW25113 strain with no vector at the same malate concentration and an elevated CO_2_ level (pCO_2_ = 0.2 atm) (Figure 3e). The evolved clone indeed showed a higher growth rate than the ancestral and wild-type strains.

We selected isogenetic populations from the 40th day of the first stage with serine and the 144th day in the second stage with malate evolution (designated as EVO-serine and EVO-malate) and sequenced their genomes using Illumina whole-genome sequencing. We identified a list of single nucleotide polymorphisms (SNPs). Appendix A shows the locations of all 29 SNPs that were found—25 from the second stage of evolution. In addition, transcriptome analysis was also performed on EVO-malate using the ancestral strain as a baseline (Appendix A). We focus our discussion on EVO-malate. Nearly half of the mutations resulted in amino acid alteration (7of 25) or were localized to a gene promotor (5 of 25). Of note, nearly half of the mutations were synonymous (13 of 25), and a significant fraction of these synonymous mutations occurred in isocitrate dehydrogenase (icd) (9 of 25). We classified the mutated genes into three broad categories, as described below. The first category consists of gene-encoding enzymes with a direct metabolic link to the function of the TCA cycle. We found one missense mutation (A398G) in the coding region of icd, close to the NADP-binding site (Arg395), and one mutation in the promotor of poxB. We surmise that the reaction catalyzed by isocitrate dehydrogenase is the bottleneck step of CO_2_ assimilation. (This is feasible because the reaction catalyzed by icd has a low standard Gibbs free energy change Δ_f_G′° = 6.4 kJ/mol (at pH = 7 and ionic concentration 0.1 M with eQuilibrator) and is hence considered to be reversible [33].) Moreover, the icd regulates the precision partitioning of carbon flux between the TCA cycle and the glyoxylate bypass, which is essential for growth when the bacterium grows on low-quality carbon sources such as acetate [34,35]. We also speculate that the mutated icd gene aims to evade such robust control on the active form of icd [34,35]. The crucial role of icd is also manifested in a high frequency of synonymous mutations among the total SNPs in icd (9 of 10), which is consistent with a growing body of evidence that synonymous mutations can be fixed in adaptive evolution [36,37,38,39]. The gene poxB encodes pyruvate dehydrogenase, which is a peripheral cell membrane enzyme that catalyzes the oxidative decarboxylation of pyruvate to form acetate and CO_2_. We hypothesize that the down regulation of poxB reduces decarboxylation and hence biomass loss due to the release of CO_2_. A second category of mutated genes consists of those commonly observed in previous adaptive laboratory evolution experiments [40]. Members of this group include rpoB (I905L) [41], rpoA [42], and dnaK [42]. The last category of mutated genes (phsC, TYMS, rlmE, nuoG, yfhM, ptsI, HigA, and tdcD) includes mutations that currently have no characterized roles. Regarding the mutations in plasmid, no mutation was found in the coding region of the two subunits of OGOR, i.e., korA and korB, but the gene expression level of subunit korB in EVO-malate was higher than the ancestral strain by a factor of 8.5 (Appendix A), which can potentially increase the enzyme capacity of OGOR and hence CO_2_ assimilation capability. This is also reasonable since the formation of OGOR from two subunits may not be optimized in the physiological conditions of *E. coli,* and hence, evolutionary tuning can further optimize such formation by increasing the concentration of one subunit. Mutation points common to both EVO-serine and EVO-malate are one SNP in the promoter of SecM and one synonymous mutation in sda in the coding region of L-serine ammonia-lyase. Finally, we briefly remark on the other two evolution experiments (56 days in serine and 146 days in malate). The mutation result is shown in Appendix A. Overall, from mutation data, the microbial population seems to follow an entirely different route in the fitness landscape. Of note, icd gene mutation was not found in these two experiments. In term of physiology, there was also no diauxic growth in the second stage using malate.

Since H_2_ is constantly used as an energy source in the evolutionary setup, it is possible for *E. coli* to metabolize H_2_ more efficiently for evolutionary fitness, but we did not observe any mutation associated with hydrogenase genes. Finally, we comment on additional results from transcriptome analysis (Appendix A). Appendix A shows that all the genes related to galactitol degradation were downregulated, because there is no demand for galactitol degradation [43]. Further, a tryptophan transporter of high affinity (mtr gene) has been upregulated. Such a high-affinity transporter is able to bind the solute at a very low concentration and can potentially pump back endogenously synthesized tryptophan that constantly leaks out [44]. Fructose-bisphosphate aldolase was also downregulated. Fructose-bisphosphate aldolase is the primary site of nickel toxicity, and the culture medium contains 0.01 mM NiCl_2_ [45]. Downregulation of fructose-bisphosphate aldolase can potentially avoid cellular toxicity due to the presence of a nickel ion.

## 4. Discussion

In this study, we have demonstrated that the combination of a recombinant expression of a key carboxylation enzyme from the rTCA cycle and laboratory evolution can rapidly evolve a strain to assimilate CO_2_. The striking ability of *E. coli* to develop a novel metabolic phenotype such as CO_2_ mixotrophy again affirmed the intricate dynamics of the actualizing and refining steps for the emergence of a novel biological phenotype [4,5,46,47]. We speculate that the existing TCA cycle has been fully reversed. The reactions catalyzing most enzymes in the existing TCA cycle in *E. coli* are reversible, and the thermodynamically unfavorable reactions are those catalyzed by citrate synthase and 2-oxogluterate dehydrogenase. First, the recombinant OGOR reverses the reaction catalyzed by 2-oxogluterate dehydrogenase. Ferredoxin has been shown to exist in *E. coli* [48], and the OGOR reaction with ferredoxin from *E. coli* has Gibbs reaction energy ΔG′° = 17.4 kJ/mol, satisfying the thermodynamic reversibility criteria, i.e., ΔG′°< 30 kJ/mol (supporting information), and this result is also consistent with the enzyme activity data from *Magnetococcus marinus* [21]. The remaining bottleneck arises from citrate synthase, which is regarded as one of the irreversible steps in the oxidative TCA cycle [19,20]. We hypothesize that the exiting citrate degradation pathway is utilized to catalyze the cleavage of citrate to acetate and oxaloacetate via citrate lyase (EC 4.1.3.6) [49,50,51]. Because the citrate molecule is very stable, it has to be activated by forming a thioester bond with the enzyme before cleavage can take place. The enzyme has three subunits, and the corresponding genes (citD, citE, and citF) are known to exist in *E. coli* from genomic analysis, and the computed Gibbs free energy is ΔG′° = 4.4 kJ/mol at pH = 7 and ion strength of 0.1 M [33]. (The enzyme activities of citrate lyase were also detected in our previous study [23].) In addition, we hypothesize that the reaction catalyzed by isocitrate dehydrogenase is the flux bottleneck for CO_2_ assimilation, which is supported by the concentration of the mutation points on this specific enzyme. (In contrast, in the case of RuBisCO-based autotrophy, the evolution targets the flux branchpoints, connecting the non-native Calvin cycle to a biosynthetic pathway, in order to satisfy the autocatalytic requirement [4,5].) In addition, we can examine the possible carbon dioxide fixation from pyruvate, a three-carbon compound. From an extensive list of carboxylase [52], we identified the relevant carboxylase candidates as pyruvate carboxylase and malic enzyme. *E. coli* lacks the gene encoding the anaplerotic enzyme pyruvate carboxylase [53]. It is also unlikely for pyruvate to be carboxylated back to malate via the malic enzyme, because net consumption of malate is known. We can therefore rule out the possibility of carbon dioxide fixation from pyruvate. It will be part of our future work to find out how CO_2_ assimilation can arise while satisfying the regulatory and network-topological requirements based on this systemized set of speculations [52,54]. One can only surmise from previous theoretical consideration that 2-oxoglutarate, as a master metabolic regulator, may play an important role [55,56].

We wish to comment on an additional control microbial growth experiment related to hydrogen dependence. We have performed an exhaustive list of control experiments to show that hydrogen indeed serves as an energy source. Of note, the evolved strain indeed shows enhanced growth due to hydrogen with malate as a carbon source and elevated CO_2_ (pCO_2_ = 0.2 atm) (Appendix A). One physiological function of prokaryotic hydrogenases is the oxidation of hydrogen gas coupled to the energy-conserving reduction of electron acceptors [57,58,59]. In *Escherichia coli*, this function is fulfilled by membrane-bound, periplasmically oriented hydrogenase isoenzymes1, 2 (Hyd-1 and Hyd-2) for H_2_-uptake according to the reaction H_2_^−^> 2H^+^ + 2e^−^ [57,58,59]. In particular, hydrogenase-1 (Hyd-1) is produced under anaerobic conditions only in the late stationary phase of cell growth, so the fact that hydrogenase-2 (Hyd-2) is able to link hydrogen oxidation to quinone reduction during anaerobic respiration is considered most relevant in this study [59].

As a caveat of our study, the emergence and disappearance of diauxic growth behavior is also interesting. These features may serve as clues regarding to how perform cell use practical heuristics and make decisions to switch to non-native CO_2_ assimilation at a regulatory level [56]. Finally, hydrogen is drawing increasing attention as a preferable “green” energy source if produced via the electrolysis of water with energy solely from a renewable source such as solar cells. Engineering *E. coli* to utilize hydrogen energy for CO_2_ assimilation is a critical milestone toward the electromicrobial conversion of CO_2_ into useful valuable chemicals using green hydrogen [60,61,62].

## Figures and Tables

**Figure 1 microorganisms-11-00253-f001:**
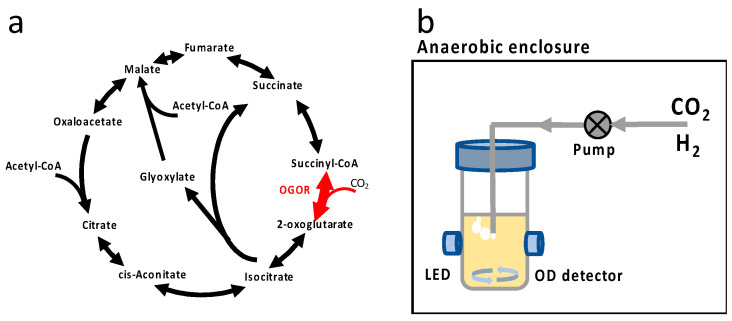
(**a**) OGOR enzyme and tricarboxylic cycle with glyoxylate bypass in *E. coli*. For reversible enzyme reactions, the enzyme is symbolized with a double-ended arrow. For OGOR reaction, one CO_2_ molecule is assimilated, and succinyl-CoA is converted to 2-oxoglutarate. (**b**) Anaerobic bioreactor for evolution. The cultivation vessel is placed in an anaerobic enclosure, and the optical density is monitored with a light-emitting diode (LED) and a photodetector. Gas pack (not shown) or N_2_ purge is used to create anaerobic conditions for elevated CO_2_ conditions and absence of CO_2_, respectively. H_2_ and CO_2_ are generated via chemical reaction and actively pumped into the culture. Growth data are wirelessly transmitted to the external computer for post-processing.

**Figure 2 microorganisms-11-00253-f002:**
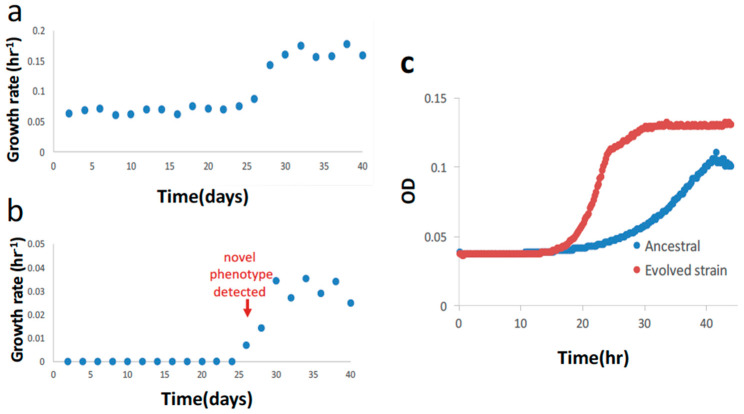
Results of first stage of evolution with serine. (**a**) Primary growth rate during the first stage of evolution with serine. (**b**) Secondary growth rate during the first stage of evolution with serine. (**c**) Growth curve of the evolved clone and ancestral strain. The evolved clone shows diauxic growth, with first phase growth rate of 0.16 hr^−1^ and second phase growth rate of 0.0251 hr^−1^. The ancestral strain shows only one growth rate of 0.06 hr^−1^. In (**a**,**b**), evolution is performed with constant serine concentration 0.5 g/L.

**Figure 3 microorganisms-11-00253-f003:**
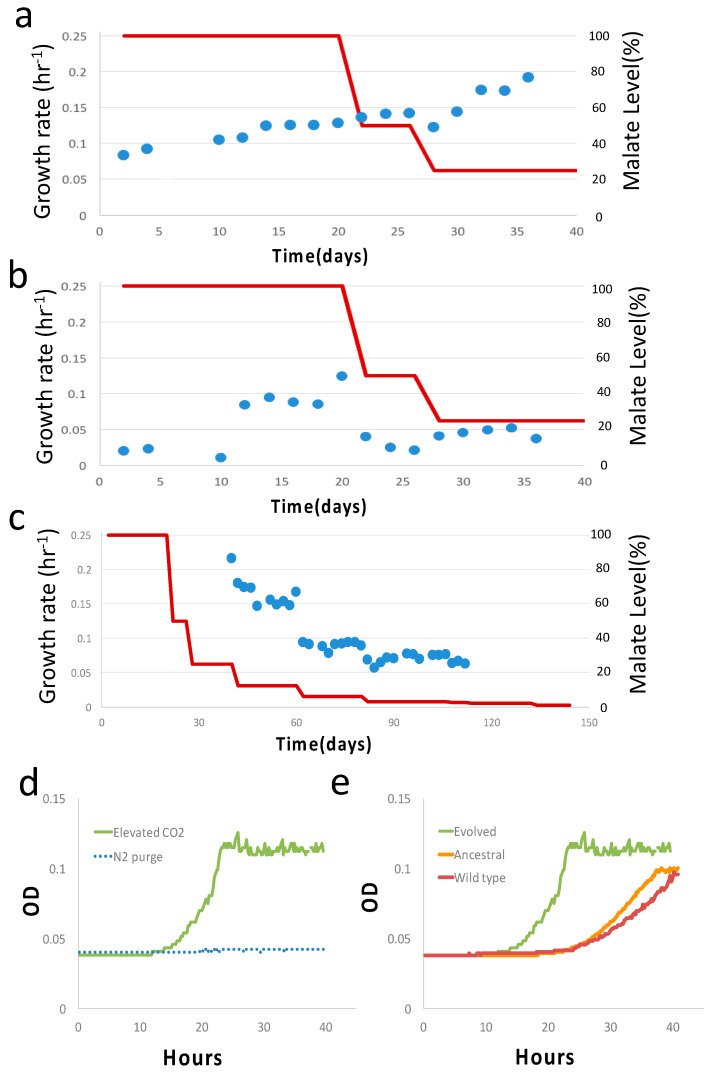
Results of the second stage of evolution with malate. (**a**) Primary growth rate of diauxic growth during the second stage of evolution with malate level (red line). (**b**) Secondary growth rate (blue dots) of diauxic growth during the second stage of evolution with malate level (red line). (**c**) Growth rate of single phase of growth from 40th day onward. Data are omitted from 112th day onward due to lack of resolution for growth rate calculation. (**d**) Growth curve of the evolved clone at the end of evolution (144th day) at elevated CO_2_ level (pCO_2_ = 0.2 atm) and no CO_2_ (N_2_ purge) at 25% malate concentration with H_2_ supply. (**e**) Growth curve of the evolved clone, ancestral strain, and control wild-type sample (BW25113 strain) at elevated CO_2_ level (pCO_2_ = 0.2 atm) and at 25% malate concentration. In all figures, 100% malate concentration corresponds to 4.9 g/L.

**Table 1 microorganisms-11-00253-t001:** The primers used to construct the expression plasmid by Ordered Gene Assembly in *Bacillus subtilis* (OGAB) method.

Primer ^a^	Primer Sequence (5′ to 3′) ^b^
KOR-DraIII-f	ATGCAC**GTT**GTGGAAAAAAGGAAGAGGGGATACCCATGAGTGAC
KOR-PflMI-r	ATGCCAA**AGA**TTGGTCAGTTGATCGTCCAGGTGCTGTTGC

^a^ DraIII and PflMI indicate the restriction enzyme used for the OGAB method. ^b^ Underlined bases represent restriction sites for DNA cloning. Bolded bases represent the three complementary nucleotides designed for the OGAB method to assemble DNA fragments.

**Table 2 microorganisms-11-00253-t002:** The enzymatic activity of 2-oxoglutarate:ferredoxin oxidoreductase.

Enzyme Activity (nmol min^−1^ mg^−1^ Protein)	
Strains	2-oxoglutarate:ferredoxin oxidoreductase
wild-type ^a^	<0.01
ancestral ^b^	49.6 ± 12.2

^a^ The wild-type strain name represents the *E. coli* BW25113 strain containing the control plasmid pGETS118. ^b^ The ancestral strain name represents the *E. coli* BW25113 strain containing the plasmid pGETS-K with 2-oxoglutarate:ferredoxin oxidoreductase genes.

## Data Availability

Data are available upon communication with the corresponding author.

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
