# Peer review of "Conversion of Escherichia coli into Mixotrophic CO2 Assimilation with Malate and Hydrogen Based on Recombinant Expression of 2-Oxoglutarate:Ferredoxin Oxidoreductase Using Adaptive Laboratory Evolution"

_microorganisms, 2023, doi:10.3390/microorganisms11020253_

Round 1

Reviewer 1 Report

Cheng et al. reported an experimental evolution study where E. coli adapted to use CO2 to grow. The work is solid, the manuscript needs extensive editing though. 

Major comments: 

  1. The Introduction is short. Experimental evolution works in chemostat and variants need to be discussed in Introduction, i.e., chemostat (doi.org/10.1128/JB.01606-14), turbidostat (doi.org/10.1093/gbe/evz197), and morbidostat (doi.org/10.1038/ng.1034). Here is a good review of chemostat (doi.org/10.1111/1574-6976.12082). 

  2. I am not sure if the authors ran into any trouble adding the trace elements into M9. But we used to suffer from this, until that we found the trace elements bottles from Corning (#25-021-CI, #25-022-CI, and #25-023-CI). They worked well for our M63 broth. Hopefully this can help save some time for the authors. 

  3. FIgures 2 and 3 are too drab. Please try to generate decent growth curve figures. 

Minor comments: 

  1. Line 29, "alternative pathway" → "an alternative pathway". 

  2. Line 32, remove "his". 

  3. Line 63, remove "the". 

  4. Line 68, "regular E. coli cloning host strain BW25113 via heat shock transformation" → "E. coli BW25113". BW25113 is the starting strain of the KEIO collection. However, I would not say it is a heavily used strain for cloning purposes. 

  5. Line 152, "suspension" → "resuspended". 

  6. Lines 156–157, "user manual" → "the manufacturer’s protocol". 

  7. Line 165, "using BWA to get basic sequence information". This does not make sense. BWA is for alignment. The basic steps are: QC → alignment → variant calling → variant annotation. Please rewrite this part and follow the logic. 

  8. Line 185, remove "Importantly". Try your best to avoid words like this with no actual meanings. 

  9. Line 205, "constant concentration 0.5 g/L" → "a constant concentration of 0.5 g/L". 

  10. Line 228, "At elevated CO2 condition" → "Under the elevated CO2 condition". 

  11. Line 249, "divide" → "classify". 

  12. Line 253, to be honest, I do not see "surmise" in a scientific paper very often. 

  13. I cannot keep building up this list. Please seek language help from professionals.

Reviewer 2 Report

The authors used adaptive laboratory evolution to develop E. coli capable of mixotrophic CO2 assimilation. Recombinant 2-oxoglutarate::ferredoxin oxidoreductase was used for CO2 assimilation, serine or malate was used as auxiliary carbon source, and hydrogen was used as energy source.

1.     Adaptive laboratory evolution experiments are often done in parallel serial cultures to identify beneficial mutations across independent replicates that can explain the improved phenotype. How many independent replicates were used in the adaptive laboratory evolution experiment? How many isolates for each evolved population were sequenced?

2.     In the adaptive laboratory evolution, the hypothesis was mutants with CO2 assimilation capability will have selective advantage. There are a few things need to be clarified for the experimental design. Why was the experiment performed in two different stages with serine and malate, and how is the growth on serine related to the growth on malate?

3.     Serial transfer is usually done during exponential growth phase when using growth as selective pressure. However, the cultures were grown for ~48 hr for each passage in this stury and the growth curves indicate that at 48 hr cells are already at stationary growth phase (Figure 2C, Figure 3C, and Figure 3D). This would result in selection for other fitness such as survival during stationary phase, and could be why there was a secondary growth phenotype. The authors should discuss the implications of this design.

4.     The Figure 3A and 3B needs clarification. Are 3A and 3B in the correct order? The primary growth vanishes on 38th day. Is it secondary growth only since then, and what does it mean? The secondary growth data were omitted from 112th day due to lack of resolution. The authors should improve the method for growth rate calculation or provide growth data from 112th day by manual calculations.

5.     In the second stage of evolution with malate, the growth rate is mostly decreasing after 38th day. Did the authors keep the frozen samples at multiple points during the adaptive evolution? Sequencing of those could help identifying causal mutations for improved growth or CO2 assimilation.

6.     It is still not clear what really caused the improved phenotype in the evolved strain. In the absence of any other evidence, the evolved phenotype needs to be explained by introducing identified mutations into the ancestral strain or reverting mutations back to wild type in the evolved strain.

Reviewer 3 Report

- Please, revise the abstract for the status of words and numbers; subcript, italic, ... etc.

- Some references are not matched with journal style.

Author Response

Thanks for suggestions.

We have made the correction in the abstract and references.

Round 2

Reviewer 1 Report

Thank you for making the revisions. I have no further questions.

Reviewer 2 Report

Thank you for addressing my comments and revising the manuscript. I have no further comments.